# Optimizing Social Network Interventions via Hypergradient-Based Recommender System Design

**Marino Kühne** [1]  **Panagiotis D. Grontas** [1]  **Giulia De Pasquale** [2]  **Giuseppe Belgioioso** [3]
**Florian Dörfler** [1]  **John Lygeros** [1]

## Abstract

Although social networks have expanded the range of ideas and information accessible to users, they are also criticized for amplifying the polarization of user opinions. Given the inherent complexity of these phenomena, existing approaches to counteract these effects typically rely on hand-crafted algorithms and heuristics. We propose an elegant solution: we act on the network weights that model user interactions on social networks (e.g., ranking of users' shared content in feeds), to optimize a performance metric (e.g., minimize polarization), while users' opinions follow the classical Friedkin-Johnsen model. Our formulation gives rise to a challenging, large-scale optimization problem with non-convex constraints, for which we develop a gradient-based algorithm. Our scheme is simple, scalable, and versatile, as it can readily integrate different, potentially non-convex, objectives. We demonstrate its merit by: (i) rapidly solving complex social network intervention problems with 4.8 million variables based on the Reddit, LiveJournal, and DBLP datasets; (ii) outperforming competing approaches in terms of both computation time and disagreement reduction.

## 1. Introduction

*Problem description and motivation:* Social media platforms brought many societal benefits but they have also been associated with the spread of fake news and the polarization of opinions (Allcott & Gentzkow, 2017; Bail et al., 2018; Bakshy et al., 2015; González-Bailón et al., 2023; Allcott et al., 2020; Levy, 2021; Guess et al., 2023a;b). The algorithms employed by social media companies to maintain user engagement are thought to contribute to the isolation of the users into echo chambers, where opinions are segregated by party lines and ties to opposing views are severed (Pariser, 2011). Even though the literature suggests that the algorithms of major social media companies may decrease the share of cross-cutting content in users' feeds (Bakshy et al., 2015; González-Bailón et al., 2023), thereby limiting diverse content exposure, the user's choice of engaging with opinion-confirming material tends to be a comparable cause of this isolation (Garrett, 2009; González-Bailón et al., 2023; Wojcieszak et al., 2022).

The field of opinion dynamics studies how the content provided by social media shapes user opinions and how opinions propagate in a social network (Noorazar, 2020). Users in the social networks are modeled as nodes in a directed, weighted graph, where the edge orientation represents the direction of influence between users and the edge weights represent the intensity of the social influence. Here we consider the well-established Friedkin-Johnsen (FJ) model (Friedkin & Johnsen, 1990), which has been successfully employed in the study of polarization and disagreement in social networks (Chitra & Musco, 2020; Musco et al., 2018; Chen et al., 2018). In the FJ model, users have both a constant internal opinion (prejudice) and a time-varying external, declared opinion. The external opinion of each user evolves by repeated averaging of the internal opinion with the external opinions of interacting users, while the prejudice remains constant. We propose a unified framework to address a broad class of problems involving the influence of user opinions on social platforms, such as polarization mitigation, through modifications to users' connections when opinions evolve according to the FJ dynamics.

*Literature review:* The literature on network interventions under the FJ dynamics is vast (Bindel et al., 2015; Wang & Kleinberg, 2023; Gionis et al., 2013; Gaitonde et al., 2020; Ancona et al., 2022; Chen et al., 2018; Musco et al., 2018; Matakos et al., 2017; Zhu et al., 2021; Chitra & Musco, 2020; Cinus et al., 2025). Most of these works seek to achieve various objectives by designing interventions on the

---

[1]ETH Zürich, Switzerland [2]Eindhoven University of Technology, The Netherlands [3]KTH Royal Institute of Technology, Sweden. Correspondence to: Marino Kühne <marino.kuehne@outlook.com>.

*Proceedings of the 42^{nd} International Conference on Machine Learning*, Vancouver, Canada. PMLR 267, 2025.

social network topology or on the internal opinions of users. The most common approach is based on adding or deleting edges from the network, where the following objectives are considered: minimization of average- and worst-case opinion dynamics metrics (Chen et al., 2018), social cost minimization (Bindel et al., 2015), and polarization and disagreement minimization (Zhu et al., 2021). Differently, (Chitra & Musco, 2020) minimizes disagreement by altering the edge weights, and (Cinus et al., 2025) minimizes various objectives by altering the edge weights without prior knowledge of internal opinions. (Wang & Kleinberg, 2023) investigates the effects of link formation on polarization and disagreement. More closely related to our work, (Cinus et al., 2023) minimizes polarization and disagreement using a projected gradient scheme. While similar in approach, our algorithm can seamlessly handle more general objectives and constraints on the weight modification. A second, less common approach focuses on changing the opinions of specific users (Gionis et al., 2013; Matakos et al., 2017; Musco et al., 2018). To achieve a generally favourable opinion about a product, (Gionis et al., 2013) sets the opinions of a fixed number of users to a specific value. On the other hand, (Matakos et al., 2017) sets the opinions of a fixed number of users to zero to minimize polarization. Finally, (Musco et al., 2018) modifies opinions to minimize the sum of polarization and disagreement.

If we view the weights[1] of the network as hyperparameters of a model, then choosing the optimal ones becomes a hyperparameter tuning problem. This defines the upper level of a bilevel optimization problem. In turn, the equilibrium resulting from the repeated averaging of the FJ dynamics can be interpreted as the Nash equilibrium of a game defined by a separate optimization problem (Bindel et al., 2015), forming the lower level. These bilevel optimization problems are difficult to solve – for example, some formulations in (Bindel et al., 2015) and (Matakos et al., 2017), are shown to be NP-hard. A few special problems with tailor-made objectives are shown to be convex, such as (Gaitonde et al., 2020, Section 4) or (Musco et al., 2018). However, this is not the case in general and the problems are solved with problem-specific algorithms or heuristics such as in (Gionis et al., 2013) or (Matakos et al., 2017).

For both large-scale convex and non-convex problems, first-order methods have proven highly effective, as demonstrated in works like (Bottou et al., 2018; Bottou, 2010; Kingma & Ba, 2015). These methods are particularly appealing due to their computational efficiency and scalability. Recent advancements have extended these approaches to hypergradients (Maclaurin et al., 2015) (the gradient of the objective with respect to hyperparameters), while also analyzing the

computational complexity of such methods (Grazzi et al., 2020). A widely-used technique for obtaining hypergradients involves differentiating through implicit mappings, as explored in (Grontas et al., 2024).

*Contributions:* We consider the FJ model of opinion dynamics. We seek to drive the equilibrium user opinions to a desired configuration by modifying network weights. Our contribution is three-fold:

1. We formulate an optimal intervention task as an optimization problem with non-convex constraints that describe the equilibria of the FJ dynamics. Our formulation can accommodate any (differentiable) objective, e.g., polarization or disagreement reduction, as well as convex constraints modelling intervention limits. Further, we can address both directed and undirected graphs, by appropriately defining the constraints.

2. We propose a first-order algorithm that alternates between modifying the network weights and computing both the equilibrium opinions and their gradient with respect to weights, by efficiently backpropagating through the FJ dynamics. Crucially, both steps are simple and scalable: they consist of a projected gradient update and the solution of two linear systems.

3. We provide a modular, and well-documented implementation of our algorithm in JAX (Bradbury et al., 2018), facilitating the usage of hardware accelerators[2]. We then test our algorithm on real-world datasets, with up to 4.8 millions modifiable weights, and obtain a solution in a matter of minutes on a GPU. Additionally, we compare our approach against the nonlinear programming solver `IPOPT`, demonstrating up to three orders of magnitude faster runtime while achieving superior solution quality. Finally, compared to an existing heuristic, which is tailored to a subclass within our problem formulation, our scheme obtains significantly better solutions.

**Notation** Let $\mathbb{R}_{\geq 0}$ ($\mathbb{R}_{>0}$) denote the set of non-negative (positive) real numbers. We denote the $(i, j)$ element of matrix $W$ by $w_{ij}$. Let $\|\cdot\|_F$ and $\|\cdot\|_2$ denote the Frobenius and the $\ell_2$ norm, respectively. The projection of a vector $w$ onto a convex set $\mathcal{W}$ is denoted by $\Pi_{\mathcal{W}}[w] = \arg\min_{z \in \mathcal{W}} \|z - w\|_2^2$. Let $F : \mathbb{R}^m \times \mathbb{R}^n \to \mathbb{R}^p$ be a vector-valued differentiable mapping. We denote the partial Jacobians of $F$ with respect to the first and second arguments as $J_1 F(w, y) \in \mathbb{R}^{p \times m}$ and $J_2 F(w, y) \in \mathbb{R}^{p \times n}$, respectively. If $p = 1$, we use $\nabla_1 F, \nabla_2 F$ to denote the partial gradients.

---

[1]The same principle holds if internal opinions are considered as hyperparameters, but it is not covered in this work.

[2]Our code is available online at `https://github.com/m-kuehne/BeeRS`

## 2. Problem Formulation

We consider social networks consisting of interacting users, where a recommendation system (referred to as *leader*) seeks to influence the users' opinions by controlling the interactions between them. This problem can be framed as a bilevel optimization problem, where the upper level involves the leader's interventions, and the lower level models the users, whose opinions are affected by the leader's decisions. Specifically, the leader aims to shift users' opinions toward a desired configuration by modifying network weights. In practice, these weights can represent various metrics—such as the ranking of users' shared content in feeds, where boosts or penalties influence visibility. This is similar to how algorithms like EdgeRank or Weighted PageRank (Wenpu & Ghorbani, 2004) adjust ranking scores based on relevance signals. Recommender systems may also weaken connection strength by reducing the frequency or prominence of shared content in users' feeds.

### 2.1. Lower Level – Opinion Dynamics

We consider a network $G$, represented by a directed weighted graph $G = (V, W)$ where $V := \{1, \ldots, n\}$ is the set of nodes, and $W \in \mathbb{R}^{n \times n}$ is the (weighted) adjacency matrix. For notational convenience, we will reshape $W$ as a vector $w \in \mathbb{R}^{n^2}$. We assume that $G$ contains no self-loops and has non-negative weights, formally, $w \in \mathcal{B} := \{w \in \mathbb{R}^{n^2} \mid w_{ii} = 0, w_{ij} \geq 0, \forall i, j \in V\}$. An edge between nodes $i, j \in V$ exists if and only if $w_{ij} > 0$, and we say that $i$ and $j$ are neighbors. Interpreting $G$ as a social network, each of the $n$ nodes corresponds to a user (e.g., individuals, companies, news outlets, etc.), and the edges represent connections between them. For each edge $(i, j)$, the user on the tail $i$ is influenced by the user on the head $j$. The edge is assigned a weight proportional to the influence $j$ exerts on $i$ and could, for example, represent the time $i$ spends consuming content shared over this connection, the frequency of interaction, or any other engagement metric. It is natural to assume that a longer exposure results in a greater influence. According to the FJ opinion dynamics (Friedkin & Johnsen, 1990), the opinion of user $i$ consists of two parts: the constant internal opinion, or prejudice, $s_i \in \mathbb{R}$ and the time-varying external opinion $y_i \in \mathbb{R}$. We consider opinions to be in the interval $[-1, 1]$, where $-1$ indicates complete opposition and $1$ complete approval. The internal opinion $s_i$ is kept private, while the external opinion $y_i$ is shared with others. Owing to the interactions between users, the external opinion of user $i$ changes over time according to the FJ update rule

$$y_i^{(t+1)} = \frac{s_i + \sum_{j \in V} w_{ij} y_j^{(t)}}{1 + \sum_{j \in V} w_{ij}}, \quad (1)$$

and we note that the sums are implicitly restricted to the neighbors of $i$, since $w_{ij} = 0$ if $j$ is not a neighbor of $i$. The update (1) can be understood as a repeated weighted averaging between the internal opinion $s_i$ and the neighbors' opinions $y_j$, where the weights $w_{ij}$ indicate how susceptible user $i$ is to user $j$'s opinion. We can compute the equilibrium of (1) as the solution of the linear system of equations

$$\underbrace{\begin{bmatrix} 1 + \sum_{j \in V} w_{1j} & \cdots & -w_{1n} \\ \vdots & \ddots & \vdots \\ -w_{n1} & \cdots & 1 + \sum_{j \in V} w_{nj} \end{bmatrix}}_{:= A(w)} y = s, \quad (2)$$

with $s = [s_1, \cdots, s_n]^\top \in \mathbb{R}^n$, $y = [y_1, \cdots, y_n]^\top \in \mathbb{R}^n$, and[3] $A(w) \in \mathbb{R}^{n \times n}$. Note that $A(w)$ depends on the network weights $w$, i.e., the vectorized form of the adjacency matrix $W$. Next, we show that whenever the weights are appropriately chosen, there exists a unique vector of equilibrium opinions.

**Proposition 2.1.** *For all $w \in \mathcal{B}$, the matrix $A(w)$ is invertible and the FJ dynamics (1) admits a unique equilibrium that solves (2).*

The proposition holds since $A(w)$ is strictly row diagonally dominant and therefore invertible (Taussky, 1949, Th. IV).

### 2.2. Upper Level – Network Intervention

The goal of the leader is to find weights $w \in \mathbb{R}^{n^2}$ that minimize a cost function $\varphi : \mathbb{R}^{n^2} \times \mathbb{R}^n \to \mathbb{R}$, mapping $w$ and the resulting solution of (2) to some upper-level objective.

**Assumption 2.2.** The function $\varphi(w, y)$ is continuously differentiable in both arguments.

A possible choice is given by $\varphi(w, y) := \sum_{i=1}^{n}(y_i - \frac{1}{n}\sum_{j=1}^{n} y_j)^2$, a classical measure of polarization, previously used by (Chitra & Musco, 2020). In practice, the leader cannot modify the network weights arbitrarily, as such interventions directly impact the users' experience (e.g., consider the case where a connection between two close friends is removed). We model permissible interventions as a constraint $w \in \mathcal{W}$, obeying the following assumption.

**Assumption 2.3.** The set of constraints $\mathcal{W}$ is non-empty, closed, and convex, and $\mathcal{W} \subseteq \mathcal{B}$.

The set $\mathcal{W}$ can be used to add constraints that encode requirements such as: (i) some connections cannot be formed or altered, by fixing $w_{ij}$ to zero or to its initial value, respectively; (ii) ensuring that some connections are maintained by

---

[3]The matrix $A(w)$ is often represented in the literature as $A(w) = L(w) + I$, where $L$ is the Laplacian matrix of the graph, and $I$ the identity matrix (Cinus et al., 2023).

imposing lower bounds on some $w_{ij}$. Further, $\mathcal{W}$ ensures that weights are compatible with our assumption $w \in \mathcal{B}$.

Note that many of these constraints decrease the number of degrees of freedom in the optimization problem, for example, by fixing some weights to a given value. In principle, one can maintain all the degrees of freedom and enforce this requirement through the set $\mathcal{W}$. Computationally, however, it is more efficient to drop the corresponding decision variables from $w$ from the beginning, giving rise to a decision vector of dimension lower than $n^2$. We always drop the weights $w_{ii}$ from $w$, as self-loops are absent, reducing the number of decision variables to $m = n(n-1)$. Throughout the remainder of this paper, $m$ denotes the effective number of decision variables.

This property of our algorithmic approach allows to further optimize for performance by deploying it on a subset of the network. One meaningful way to choose which weights to optimize for (or which areas of the networks to affect) would be to use heuristics to quantify the importance of nodes, and then optimize the weights of edges connected to these nodes. For instance, we could use the degree of a node or, more generally, other centrality measures (Bloch et al., 2023).

Combining the upper-level objective $\varphi$ with the equilibrium of the lower level (2) and permissible interventions in $\mathcal{W}$, we formulate the following optimization problem:

$$\underset{w, y}{\text{minimize}} \quad \varphi(w, y) \tag{3a}$$

$$\text{subject to} \quad A(w)y = s, \tag{3b}$$

$$w \in \mathcal{W}. \tag{3c}$$

The bilinear constraint (3b) renders the feasible set non-convex. We rewrite (3) as

$$\underset{w}{\text{minimize}} \quad \varphi(w, y^\star(w)) \tag{4a}$$

$$\text{subject to} \quad w \in \mathcal{W}, \tag{4b}$$

where $y^\star(w)$ denotes the solution to the system $A(w)y = s$. We refer to $y^\star(\cdot) : \mathbb{R}^m \to \mathbb{R}^n$ as the equilibrium mapping, which is well-posed under Assumption 2.3. Formulation (4) is favorable since $y^\star(\cdot)$ is continuously differentiable (see Proposition 3.1), hence motivating us to deploy a first-order solution method.

Although (4) involves convex constraints, it is still challenging to solve, as $y^\star(\cdot)$ is an implicit mapping, for which a closed-form expression is, in general, not available. Moreover, the objective is non-convex.

Note that, in (4a) we optimize solely over the weights $w$ and not the innate opinions $s$. Our algorithm can readily be extended to jointly optimize for $w$ and $s$, which we consider as an interesting future work.

## 3. Algorithm Design

To solve (4), we propose a gradient-based algorithm. The main challenge lies in obtaining the *hypergradient*, i.e., the gradient of the objective $\varphi(w, y^\star(w))$ with respect to $w$. This can be achieved by using the chain rule as follows:

$$\begin{aligned} \nabla_w \varphi(w, y^\star(w)) = & \nabla_1 \varphi(w, y^\star(w)) \\ & + Jy^\star(w)^\top \nabla_2 \varphi(w, y^\star(w)). \end{aligned} \tag{5}$$

Under Assumption 2.2, the partial gradients $\nabla_1 \varphi$ and $\nabla_2 \varphi$ are well-defined. However, ensuring the existence of the Jacobian of the equilibrium mapping, $Jy^\star(w)$, also called the *sensitivity*, and computing it is a non-trivial task.

To compute $Jy^\star(w)$, we introduce the auxiliary expression $F(w, y) := A(w)y - s$. In the following we provide an explicit expression for the sensitivity.

**Proposition 3.1.** *Under Assumption 2.3, the function $y^\star(\cdot)$ is continuously differentiable and its Jacobian $Jy^\star(w)$ is well-defined. Moreover, $Jy^\star(w)$ is given by*

$$Jy^\star(w) = -J_2 F(w, y^\star(w))^{-1} J_1 F(w, y^\star(w)). \tag{6}$$

*Proof.* From the definition $F(w, y) := A(w)y - s$, it immediately follows that $J_2 F(w, y) = A(w)$, which is continuous in both arguments. Furthermore, $J_1 F(w, y)$ is continuous in both arguments by Lemma A.1 in Appendix A.1. Thus, $F(w, y)$ is continuously differentiable. Additionally, by Proposition 2.1, $A(w)$ and therefore $J_2 F(w, y)$ are nonsingular. By invoking the implicit function theorem (Dontchev & Rockafellar, 2009, Th. 1B.1), the sensitivity exists, is continuous, and given by (6). □

Given $Jy^\star(w)$, we can evaluate the hypergradient and solve (4) using our preferred first-order method. In this work, we deploy projected gradient descent with momentum (Bolduc et al., 2017) given by

$$\begin{aligned} m^{(k+1)} &= \gamma m^{(k)} + \nabla_w \varphi\big(w^{(k)}, y^\star\big(w^{(k)}\big)\big) \\ w^{(k+1)} &= \Pi_{\mathcal{W}}\big[w^{(k)} - \alpha^{(k)} m^{(k+1)}\big], \end{aligned} \tag{7}$$

where $\alpha^{(k)} \in [0, 1]$ is a possibly iteration-dependent stepsize and $\gamma \geq 0$ is a momentum parameter. This formulation contains a short-term memory for the descent direction of the previous iteration $k$ in the form of $m^{(k)}$, and it can often significantly decrease the runtime for an appropriate choice of $\gamma$ (Goh, 2017). The projection is employed to ensure feasibility of the iterates $w^{(k)}$.

### 3.1. Backpropagation through Equilibrium Opinions

Computing $Jy^\star(w)$ by solving the matrix equation (6) can be computationally challenging for large-scale problems

**Algorithm 1** BeeRS

1: **Input:** Tolerance $\varepsilon$, step size $\alpha^{(k)}$, momentum param. $\gamma$
2: $w^{(0)} \leftarrow$ Set initial network weights
3: **while** True **do**
4:     Compute $J_2F(w^{(k)})$
5:     $y^\star(w^{(k)}) \leftarrow$ Solve (2)
6:     **if** $\zeta^{(k)} \leq \varepsilon$ **then** Break
7:     Compute $J_1F(w^{(k)}, y^\star(w^{(k)}))$
8:     Compute gradients $\nabla_{1,2}\varphi(w^{(k)}, y^\star(w^{(k)}))$
9:     $v^{(k)} \leftarrow$ Solve (9)
10:    $\nabla_w\varphi(w, y^\star(w)) \leftarrow$ Compute (8)
11:    $w^{(k+1)} \leftarrow$ Using (7)
12: **end while**
13: **Output:** $w^\star, \varphi(w^\star, y^\star(w^\star))$

---

(Parise & Ozdaglar, 2021). Concretely, we need to solve

$$\underbrace{J_2F(w, y^\star(w))}_{\mathbb{R}^{n\times n}} \underbrace{Jy^\star(w)}_{\mathbb{R}^{n\times m}} = -\underbrace{J_1F(w, y^\star(w))}_{\mathbb{R}^{n\times m}},$$

for a given $w$, and therefore fixed $y^\star(w)$, with matrix sizes as indicated. To reduce the computational burden, note that we do not need the sensitivity $Jy^\star(w)$ by itself in (5) but rather the vector-Jacobian product $Jy^\star(w)^\top \nabla_2\varphi(w, y^\star(w))$, see e.g., (Grazzi et al., 2020). Substituting (6) in (5), yields:

$$\nabla_w\varphi(w, y^\star(w)) = \nabla_1\varphi(w, y^\star(w)) - J_1F(w, y^\star(w))^\top$$
$$\left(J_2F(w, y^\star(w))^\top\right)^{-1}\nabla_2\varphi(w, y^\star(w))$$
$$= \nabla_1\varphi(w, y^\star(w)) - J_1F(w, y^\star(w))^\top v, \quad (8)$$

where $v \in \mathbb{R}^n$ solves the linear system of equations

$$J_2F(w, y^\star(w))^\top v = \nabla_2\varphi(w, y^\star(w)). \quad (9)$$

Hence, we can compute the hypergradient by solving (9) instead of (6), i.e., we solve one linear system instead of $m$.

Combining the equilibrium backpropagation (9) with the chain rule (8) yields Algorithm 1, which we refer to as BeeRS (Best Intervention for Recommender Systems). As a termination criterion, we use

$$\zeta^{(k)} := \frac{|\varphi(w^{(k-1)}, y^\star(w^{(k-1)})) - \varphi(w^{(k)}, y^\star(w^{(k)}))|}{|\varphi(w^{(k-1)}, y^\star(w^{(k-1)}))|} \leq \varepsilon$$

for some small tolerance $\varepsilon > 0$. Implementation details are given in Appendix A.2.

# 4. Numerical Simulations & Comparisons

In this section, we demonstrate the effectiveness of our algorithm by deploying it on real-world and artificially-generated datasets. In Section 4.1, we study the scalability

of BeeRS on both a CPU and a GPU and the evolution of $\varphi$ across iterations. In Section 4.2, we compare BeeRS to IPOPT, a state-of-the-art nonlinear programming solver that can directly solve the non-convex program (3), while in Section 4.3 we compare against the algorithm proposed in (Chitra & Musco, 2020). Hardware specifications are given in Appendix A.3, and the choices for the step size $\alpha$ and momentum parameter $\gamma$ are justified in Appendix A.4.

## 4.1. Scalability Study

### 4.1.1. PROBLEM SETUP

We consider a social network with directed connections. Let us consider the problem where a newly founded news agency, labelled as $\zeta$, wants to establish itself in the network by sharing content such as news articles. The news agency is neutral, which is encoded by the internal opinion $s_\zeta = 0$, while all other users (readers) might be biased with internal opinions $s_i \in [-1, 1]$. The network operator (the leader) has to decide how to deliver the additional content to the existing users, with the goal of minimizing polarization in the network. Additionally, the operator charges a fee to the news agency for distributing its content, which is proportional to the amount of content shared over the network. This expense can be viewed as an advertisement fee, reflecting the fact that getting space on the platform is costly. The agency's advertising budget is bounded by $b \in \mathbb{R}_{>0}$.

To model this, we introduce a new connection from every existing user $i$ in the network to the news agency $\zeta$. Since the network is directed and the news agency does not have any outgoing connections, it remains neutral, hence $y_\zeta = s_\zeta = 0$. The connections from users to the news agency carry a weight $w_{i\zeta} \geq 0$, which is proportional to the share of $\zeta$'s content in user $i$'s feed. The goal of the leader is to compute weights that minimize polarization, measured as $\sum_{i=1}^n (y_i - \frac{1}{n}\sum_{j=1}^n y_j)^2$, under the news agency's budget constraint $\sum_{i=1}^n w_{i\zeta} \leq b$.

We can formalize this problem as

$$\underset{y\in\mathbb{R}^n, w\in\mathcal{B}}{\text{minimize}} \quad \sum_{i=1}^n (y_i - \frac{1}{n}\sum_{j=1}^n y_j)^2 \quad (10a)$$

$$\text{subject to} \quad A(w)y = s, \quad (10b)$$

$$w_{ij} = w_{ij}^{(0)}, \forall i, j \in V \setminus \zeta, \quad (10c)$$

$$w_{\zeta j} = 0, \forall j \in V, \quad (10d)$$

$$\sum_{i=1}^n w_{i\zeta} \leq b, \quad (10e)$$

where the variance objective is a standard polarization metric (Chitra & Musco, 2020), (10b) ensures that opinions are in equilibrium and conform to the FJ dynamics, (10c) fixes the weights that are unrelated to the news agency to their original values $w^{(0)}$, (10d) ensures $\zeta$ has no outgoing connections, and (10e) imposes the budget limit of $\zeta$.

We stress that weights that are fixed by constraints, as in (10c) and (10d), are shown for clarity of exposition. In our numerical implementation, we remove these weights to reduce the dimensionality of the problem. Therefore, (10) ends up with $m = n - 1$ decision variables for the weights.

### 4.1.2. NUMERICAL RESULTS

We solve problem (10) on two datasets: the DBLP dataset (Tang et al., 2008) and the LiveJournal social network dataset (Backstrom et al., 2006; Leskovec et al., 2008; Leskovec & Krevl, 2014), one of the largest directed social networks on the SNAP platform (Leskovec & Sosič, 2016). The DBLP dataset describes citation relationships between scientific articles: Nodes represent articles, and edges indicate citations between them, where the weight on the edge from node $i$ to node $j$ is defined as $w_{ij} = 1$ if article $i$ cites article $j$, and $w_{ij} = 0$ otherwise. By the nature of the dataset, there are no self-loops (a paper does not cite itself). We utilize the dataset topology by giving it a social network interpretation. Specifically, we reinterpret the nodes as users and the edges as a "following" relation: user $i$ follows user $j$ if and only if $w_{ij} > 0$. In this context, content shared by user $j$ appears in user $i$'s feed. The LiveJournal dataset already contains the topology of a social network. We assign to both datasets internal opinions uniformly at random from the set $\{-1, 1\}$.

We consider (10) for a varying number of users $n$ to study the computational cost of evaluating the hypergradient. To do so, we take the first $n$ nodes and the corresponding internal opinions. We set $b = \frac{n}{10}$, $\alpha = 0.05$, and $\gamma = 0.6$. We execute 1 iteration of BeeRS for every problem size 10 times on the CPU and on the GPU. We report mean runtime $\pm 1$ standard deviation over 10 runs in Figure 1.

### 4.1.3. SOLVING ON THE ENTIRE DATASET

Given the remarkable scalability of BeeRS on the GPU, we deploy it on the entire DBLP dataset consisting of $n = 3,079,007$ nodes and $25,166,994$ edges and on the entire LiveJournal dataset consisting of $n = 4,847,571$ nodes and $68,993,773$ edges. We solve (10) ten times with $b = \frac{n}{10}$, $\alpha = 0.05$, $\gamma = 0.6$, and $\varepsilon = $ 1e-2 on the GPU. BeeRS achieves a polarization reduction of 39.5% and 40.0% on the DBLP and LiveJournal dataset, respectively. The respective mean runtime until convergence for DBLP and LiveJournal is 195.4 s and 766.8 s with standard deviation of 9.6 s and 5.4 s.

### 4.1.4. ANALYSIS OF THE COST EVOLUTION

We study the evolution of opinion polarization, i.e., the value of the objective function $\varphi$ in problem (10), across algorithm iterations. We simulate problem (10) 5 times for each of the DBLP, LiveJournal, and Reddit datasets (introduced in Section 4.3) with 5 different initializations. We run each simulation for 25 iterations and display the average cost as well as $\pm 1$ standard deviation as a function of the iteration $k$ in Figure 2. We set $\alpha = 20$, $\gamma = 0.6$ for the Reddit dataset and $\alpha = 0.05$, $\gamma = 0.6$ for DBLP and LiveJournal. For all datasets we set $b = \frac{n}{10}$.

We observe that the cost consistently decreases across all three datasets, with the largest reduction occurring within the first few iterations. Further, we note that BeeRS converges in all cases in roughly 20 iterations.

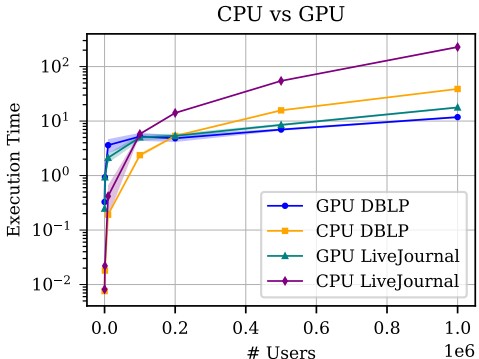

*Figure 1.* The mean runtime and $\pm 1$ standard deviation over 10 runs of 1 iteration of BeeRS on both a GPU and a CPU.

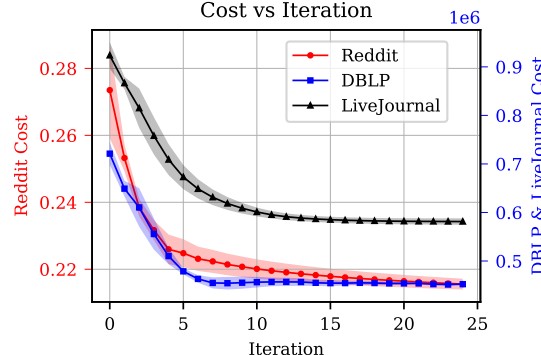

*Figure 2.* The average cost $\varphi \pm 1$ standard deviation as a function of the iteration $k$. The cost is averaged over 5 different initializations of the edge weights $w_{i\zeta}$.

We observe that for small numbers of users, the CPU implementation has a lower per-iteration runtime. This is not surprising, given the additional overhead produced by moving computations to the GPU and by JAX's just-in-time compilation, among other reasons. However, the GPU implementation becomes faster for more than 100,000 users.

## 4.2. Comparison with `IPOPT`

We now consider the problem setup from Section 4.1 and solve problem (10) using the same datasets with `IPOPT`.

### 4.2.1. NUMERICAL RESULTS

We compare the CPU implementation of BeeRS to the nonlinear solver `IPOPT` (Wächter & Biegler, 2006), using the cyipopt (cyipopt) wrapper for Python. For `IPOPT`, we use a tolerance of 1e-3 and constraint-violation tolerance of 1e-4. For BeeRS, we use $\varepsilon = $ 1e-3, $\gamma = 0.6$, and $\alpha = 0.05$. We solve (10) with both methods for a varying number of users $n$, and repeat each simulation 10 times. We set $b = \frac{n}{10}$.

We report the average runtime $\pm 1$ standard deviation for different problem sizes for both approaches in Figure 3. Note that the number of nodes considered here is significantly smaller than in Figure 1 due to the computational limitations of `IPOPT`. We observe that BeeRS outperforms `IPOPT` for all simulated $n$ on both datasets, and by a larger margin for larger $n$. The runtime difference ranges between roughly 1 and 3 orders of magnitude for the largest problems.

In terms of solution quality, BeeRS attains the lowest cost in all cases, whereas the `IPOPT` implementation exhibits relative suboptimality[4] of more than 5e-3 on the DBLP dataset and of at least 6e-2 on the LiveJournal dataset. Detailed results are reported in Appendix A.5.

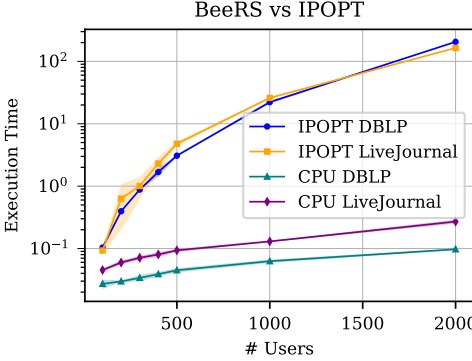

*Figure 3.* The mean runtime and $\pm 1$ standard deviation over 10 runs of BeeRS and `IPOPT` (until convergence) on a CPU.

## 4.3. Comparison with the Literature

Next, we compare BeeRS to the algorithm in (Chitra & Musco, 2020). There, the authors propose an extension to the FJ dynamics, called NAD (Network Administrator Dynamics), to highlight the potential pitfalls of recom-

---

[4]We define the relative suboptimality as $\frac{|\varphi - \varphi^\star|}{|\varphi^\star|}$, where $\varphi^\star$ is the best solution achieved by any of the two approaches.

mender systems that naively minimize disagreement, defined as $D(w, y) := \frac{1}{2} \sum_{i=1}^{n} \sum_{j=1}^{n} w_{ij}(y_i - y_j)^2$. In particular, they investigate the change in polarization in the network before and after modifying the weights to minimize disagreement using NAD. In (Chitra & Musco, 2020), the authors raise the concern that attempting to minimize disagreement, e.g., through NAD, can have adverse effects on polarization (and even disagreement itself). In our following studies, we provide some insights on this issue, and show how BeeRS circumvents it.

### 4.3.1. PROBLEM SETUP

The NAD follows an iterative approach, as summarized in Algorithm 2 in Appendix A.6. In a first step, equilibrium opinions $y^\star(w^{(k)})$ are computed based on the current weights $w^{(k)}$. In a second step, disagreement $D(w, y^\star(w^{(k)}))$ is minimized by tweaking the weights $w$ under the assumption that the opinions remain unaffected by the new weights and under convex constraints. In particular, the modified network should be similar to the original one, and the out-degree of each node should be preserved. The second step is formalized as the convex program

$$\underset{w^{(k+1)} \in \mathcal{B}}{\text{minimize}} \quad D(w^{(k+1)}, y^\star(w^{(k)})) \tag{11a}$$

$$\text{subject to} \quad \sum_{j=1}^{n}(w_{ij}^{(k+1)} - w_{ij}^{(0)}) = 0, \forall i \in V, \tag{11b}$$

$$\|W^{(k+1)} - W^{(0)}\|_F \leq \delta \|W^{(0)}\|_F, \tag{11c}$$

where $\delta > 0$ is a parameter and $W^{(0)}$ denotes the adjacency matrix of the original network with weights $w_{ij}^{(0)}$. Intuitively, constraint (11c) ensures that users of the social network do not perceive major differences in shared content and (11b) guarantees that they spend constant time on the network. The iterations continue until $w^{(k)}$ converges.

(Chitra & Musco, 2020) note that in certain cases NAD fails to reduce disagreement and polarization $P(y) := \sum_{i=1}^{n}(y_i - \frac{1}{n}\sum_{j=1}^{n} y_j)^2$ increases. To mitigate the problem, they propose adding a regularization term $\lambda \|W\|_F^2$ with a parameter $\lambda > 0$ to the objective (11a), but the method still struggles to reduce disagreement. We aim to improve upon these results.

We observe that BeeRS can be deployed on (11), without the need to assume opinions are fixed during the optimization iterations. With BeeRS, we can directly estimate the effects of a weight change on the equilibrium opinions (and therefore the effect on the objective) through the sensitivity $Jy^\star(w)$. We readapt the original problem as

$$\underset{y \in \mathbb{R}^n, w \in \mathcal{B}}{\text{minimize}} \quad \frac{1}{2} \sum_{i=1}^{n} \sum_{j=1}^{n} w_{ij}(y_i - y_j)^2 \tag{12a}$$

$$\text{subject to} \quad (11b) \text{ and } (11c), \tag{12b}$$

$$A(w)y = s. \tag{12c}$$

We add constraint (12c) to take into account the dynamic opinions $y$. We use $\delta = 0.2$ for all subsequent results.

### 4.3.2. CASE STUDY: TOY EXAMPLE

To better understand how the two optimization strategies compare with each other, we use a minimal network that allows for disagreement. It consists of two nodes with internal opinions $s_0 = 1$ and $s_1 = 0$, connected by an undirected[5] edge with initial weight $w^{(0)} = 1$. In this network, we aim to solve (12) without constraint (11b) (as enforcing this constraint would restrict the allowed weight to $w = w^{(0)} = 1$). To evaluate the behavior of NAD we first compute the resulting equilibrium opinions by solving (2) for appropriate $A(w)$ (line 2). We find $y_0^\star(w) = \frac{2}{3}$ and $y_1^\star(w) = \frac{1}{3}$. Disagreement is given by $D(w, y^\star(w)) = 0.11$. In a second step, NAD minimizes disagreement by keeping the opinions constant (line 3). Recalling the definition of disagreement, we see that $D(w, y)$ decreases for fixed $y$ if and only if $w$ is decreased. Therefore, NAD decreases $w$, in order to minimize disagreement. Then, NAD proceeds by computing the equilibrium opinions under the updated weight $w^{(k+1)}$, followed by a reduction of the edge weight $w$ to minimize disagreement for fixed opinions in a second step. It follows that the edge weight is decreased until constraint (11c) is tight, which, for $\delta = 0.2$, occurs when $w = 0.8$.

On the other hand, BeeRS updates the weight in the direction of steepest descent, taking into account the effect on opinions. By deploying it on such a simple problem, we observe that, as opposed to NAD, BeeRS *increases* the weight to $w = 1.2$. To compare the performance of BeeRS and NAD in terms of disagreement, we compute equilibrium opinions for all $w \in [0, 2]$ by solving (2), and we illustrate the resulting disagreement in Figure 4. We mark the solution of both algorithms, and we observe that BeeRS outperforms NAD. We argue that the reason for this discrepancy comes from the fact that our algorithm is able to anticipate the effect of changing weights on the equilibrium opinions and, hence, move in a direction that minimizes disagreement. By contrast, NAD myopically optimizes weights, neglecting the dynamics of opinions, and thus exacerbates disagreement.

### 4.3.3. NUMERICAL RESULTS: ADDRESSING A REAL-WORLD PROBLEM

Next, we deploy both algorithms on real-world data. We use the undirected Reddit dataset (De et al., 2014), where external opinions $y$ were extracted by sentiment analysis. Analogously to (Chitra & Musco, 2020), we set the internal opinions[6] $s \in [0, 1]$ by computing $s = A(w)y$. The dataset

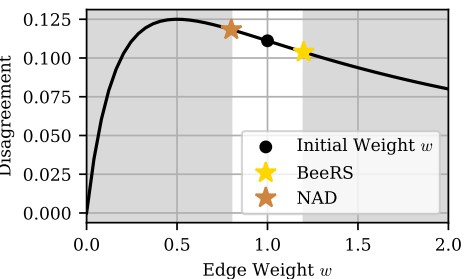

*Figure 4.* Behavior of BeeRS and NAD (without regularization) in a toy example network. We plot disagreement as a function of weights $w$ for internal opinions $s_0 = 1$ and $s_1 = 0$. The feasible region is shown in white.

*Table 1.* Comparison between BeeRS, NAD, and NAD$^\star$ with $\lambda = 0.2$. The runtime is averaged over 10 executions.

|  | BEERS | NAD | NAD$^\star$ |
|---|---|---|---|
| MEAN RUNTIME, S | 141.6 | 35.9 | 27.3 |
| MIN. RUNTIME, S | 137.7 | 34.2 | 25.8 |
| MAX. RUNTIME, S | 150.4 | 38.5 | 28.3 |
| CHANGE IN POL. $P$ | −40.2% | +128.7% | −0.7% |
| CHANGE IN DISAG. $D$ | −21.5% | +41.4% | +0.3% |

consists of 556 nodes and 8969 edges. We remove 3 disconnected users, leading to a total of 553 users. The resulting problem has $m = n(n - 1) = 305,256$ upper-level variables. Since the edges are undirected, we set half of the variables through a constraint, as shown in the toy example.

We deploy BeeRS on a CPU[7] with $\varepsilon = 1\mathrm{e}\text{-}2$, $\gamma = 0.6$, and $\alpha = 1000$, (see Appendix A.4.1). We use the default implementation of NAD from (Chitra, 2019), making minor modifications to adapt the code to our specific problem. We run both the default NAD and NAD with regularization $\lambda = 0.2$, referred to as NAD$^\star$. We summarize the runtime as well as the change in polarization and disagreement of each algorithm in Table 1. We observe substantial differences in performance among the three algorithms.

Contrary to (Chitra & Musco, 2020), BeeRS achieves a decrease in disagreement: the objective of NAD and problem (12). Further, the effect on polarization is different, as BeeRS leads to a significant decrease of polarization, whereas NAD without regularization increases polarization. Although NAD$^\star$ improves on this aspect, it is still outperformed by BeeRS in both disagreement and polarization reduction. However, the superior performance of BeeRS

---

[5]We model the undirected edge weight $w$ with the two variables $w_{01}$ and $w_{10}$, and the constraint $w_{01} = w_{10}$.

[6](Chitra & Musco, 2020) use $s \in [-1, 1]$ in their formulation. However, consistent with their implementation (Chitra, 2019), we

adopt $s \in [0, 1]$.

[7]The current implementation of BeeRS does not support the Frobenius-norm constraint on the GPU.

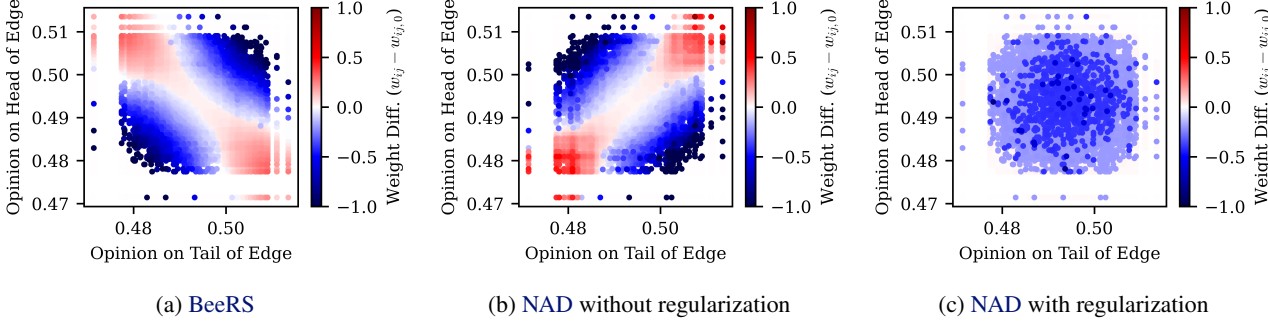

*Figure 5.* Optimal edge weight modification in relation to external opinions connected by edge computed by three approaches. The horizontal and vertical axes show the external opinion before intervention at the tail and head of an edge, respectively. The color encodes the change of edge weight during the intervention, i.e., $w_{ij} - w_{ij,0}^{(0)}$.

comes at the price of an increased runtime.

Next, we study the performance mismatch between BeeRS and NAD. To do so, we visualize the intervention mechanism, i.e., the change of weights $w_{ij} - w_{ij}^{(0)}$, in Figures 5a, 5b, and 5c for BeeRS, NAD, and NAD$^\star$, respectively.

All figures show the external opinion of users before the intervention on the horizontal and vertical axis. The x- and y-coordinates of each point represent the opinion of the user on the tail and head of the edge, respectively. The colour of each point encodes the intervention, i.e., the weight change of the corresponding edge. Since we consider an undirected network, the influence over edges is bidirectional and the plots are symmetric. As we can modify the weight of all edges (even the ones not present in the initial configuration), there are roughly 300,000 points to be plotted. To enhance the interpretability of the figures, we order all interventions by increasing absolute value and plot them increasingly on top of each other, i.e., the largest weight changes are drawn on top and are therefore visible. Note that, a large number of edges are not visible as they undergo no change in weight.

In contrast to Figure 5a and 5b, we cannot identify any obvious intervention for NAD$^\star$ in Figure 5c. The intervention seems to lack a specific pattern, although we note that many edges experience a large decrease in weight (dark blue dots) and none experience a large increase. Hence, we subsequently focus on BeeRS and NAD without regularization.

Notably, BeeRS and NAD seem to behave exactly opposite. BeeRS mainly reinforces connections between above- and below-average opinions and weakens edges connecting opinions of the same bias. Conversely, NAD strengthens edges between similar opinions and weakens the ones between opposing views.

This behavior aligns with the observations in Section 4.3.2 and explains the superior performance of BeeRS in minimizing disagreement and polarization. The intervention mecha-

nism followed by NAD does not encapsulate the fact that, when we connect opposing opinions under the FJ dynamics, more moderate opinions are formed through averaging, a favorable phenomenon in terms of minimizing disagreement and polarization. This is in line with the weight update rule of NAD (line 3) that ignores the dynamics. Instead, the backpropagation through equilibrium opinions step of BeeRS is able to explicitly incorporate and exploit this effect.

## 5. Conclusion and Future Work

We formulated the task of finding the best network intervention given any differentiable objective function under FJ dynamics as an optimization problem with non-convex constraints, and we solved it with a gradient-based algorithm. We demonstrated the flexibility and scalability of our method by successfully deploying it on various optimization problems of different sizes. Our algorithm significantly outperforms a state-of-the-art solver in terms of runtime, while achieving the same solution quality.

As future work, we would like to make our approach more practical by connecting the weight modifications to actionable interventions in network, such as boost or penalty factors in a weighted PageRank fashion (Wenpu & Ghorbani, 2004). Further, we envision increasing the scalability of our method in two ways: (i) updating only a (potentially randomly-chosen) subset of the modifiable weights at each iteration, similar to coordinate descent methods (Wright, 2015), and (ii) solving the linear systems (2) and (9) in a distributed fashion across multiple devices (CPUs or GPUs).

One limitation of our algorithm is the convexity assumption on $\mathcal{W}$, which precludes the use of binary decision variables: is an edge present or not? Our framework could incorporate binary decisions by relaxing the binary constraints or employing heuristics. Preliminary numerical experiments indicate this to be a promising future research direction.

## Acknowledgements

This work was supported by the Swiss National Science Foundation under the NCCR Automation (grant agreement 51NF40_225155), and by the Wallenberg AI, Autonomous Systems and Software Program (WASP) funded by the Knut and Alice Wallenberg Foundation.

## Impact Statement

This paper presents work whose goal is to advance optimization methods for recommender systems. There are many potential societal consequences of our work, none of which we feel must be specifically highlighted here.

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

# A. Appendix

## A.1. Computation of $J_1 F(w, y)$

Combining the definition of $A(w)$ in (2) with the definition $F(w, y) := A(w)y - s$, we arrive at

$$F(w, y) = \underbrace{\begin{bmatrix} 1 + \sum_{j \in V} w_{ij} & -w_{12} & \cdots & -w_{1n} \\ \vdots & & \ddots & \vdots \\ -w_{n1} & -w_{n2} & \cdots & 1 + \sum_{j \in V} w_{ij} \end{bmatrix}}_{A(w)} y - s. \tag{13}$$

Now we can compute $J_1 F(w, y)$, the partial Jacobian of $F(w, y)$ with respect to $w$, and show that it is continuous in both arguments. Recall the definition $w := [w_{12}, \ldots, w_{1n}, w_{21}, \ldots, w_{2n}, \ldots, w_{n1}, \ldots w_{n(n-1)}]^\top \in \mathbb{R}^m$ with $m = n(n-1)$ (there are no self-loops, thus we drop the entries $w_{ii}$ for computational efficiency).

**Lemma A.1.** *The partial Jacobian of $F(w, y)$ with respect to $w$, i.e., $J_1 F(w, y)$, is continuous in both arguments.*

*Proof.* Considering (13), it is easy to see that $J_1 F(w, y)$ is given by

$$J_1 F(w, y) = \begin{bmatrix} y_1 - y_2 & y_1 - y_3 & \cdots & y_1 - y_n & 0 & \cdots & 0 & 0 & 0 & \cdots & 0 \\ & & & & & \vdots & & & & & \\ 0 & 0 & \cdots & 0 & 0 & \cdots & 0 & y_n - y_1 & y_n - y_2 & \cdots & y_n - y_{n-1} \end{bmatrix} \in \mathbb{R}^{n \times m}, \tag{14}$$

which is continuous. $\qquad\square$

## A.2. Implementation Details

We implement the algorithm in Python 3.12. We provide both a GPU- and a CPU-compatible implementation. The GPU version performs most computations with JAX (Bradbury et al., 2018) and has some NumPy (Harris et al., 2020) dependencies. We solve the linear systems (2) and (9) with a jax.scipy implementation of the GMRES algorithm (Saad & Schultz, 1986). For the projection, we use the jaxopt (Blondel et al., 2022) BoxOSQP solver. JAX functions are just-in-time compiled if applicable. We represent the adjacency matrix of the network and the Jacobians as sparse csr_array, which allows efficient handling of large datasets. The upper-level gradients $\nabla_{1,2}\varphi$ are computed with the automatic differentiation (autodiff) functionalities of JAX.

The CPU version is based on NumPy. Unlike the GPU implementation, we allow both sparse and dense representations of the network's adjacency matrix and the Jacobians. For dense matrices, we rely on NumPy to solve the linear systems (2) and (9), for sparse matrices, we use a SciPy (Virtanen et al., 2020) implementation of the GMRES algorithm. In both cases, we use CVXpy (Diamond & Boyd, 2016; Agrawal et al., 2018) to perform the projections with the following solvers: Clarabel (Goulart & Chen, 2024) for quadratic programs, and SCS (O'Donoghue et al., 2016) for second-order cone programs. The upper-level gradients $\nabla_{1,2}\varphi$ are computed with the automatic differentiation (autodiff) functionalities of PyTorch (Paszke et al., 2019).

## A.3. Hardware Specifications

Experiments were conducted on a Windows 11 machine using Windows Subsystem for Linux (WSL) version 2, either on an Intel® Core™i7-11700F CPU at 2.50 GHz with 8 cores and 16 logical processors and 32 GB of RAM or on a NVIDIA GeForce RTX 3060 Ti GPU, with 8 GB of GDDR6 video memory, running CUDA 12.6 and cuDNN 9.5.1.

## A.4. Choosing the Hyperparameters – Ablation Study

As discussed in Section 3, BeeRS employs projected gradient descent with momentum (7). This method uses two hyperparameters, namely step size $\alpha$ and momentum parameter $\gamma$. We conduct an ablation study to find parameters leading to satisfying results in terms of runtime and minimum cost.

We consider the same setup as in Section 4.1, i.e., problem (10) on the DBLP dataset version 10 (Tang et al., 2008). We consider the first $n = 1000$ users and set $b = 100$. We initialize all variables, i.e., all connections to the news agency, with the value 0. We use a tolerance of $\varepsilon = 1\text{e-}3$.

We now solve this problem with BeeRS. We grid the hyperparameters for values of $\alpha \in (0, 500]$ and $\gamma \in [0, 0.95]$. To enhance visibility, we display only some of the grid points in Figure 6, in particular $\alpha \in \{0.001, 0.002, \ldots, 0.05, 0.06, \ldots, 0.1, 0.2, \ldots, 1\}$ and $\gamma \in \{0.0, 0.3, 0.6, 0.9\}$. In the left subplot, we show the number of iterations until convergence on the vertical axis as a function of $\alpha$ on the horizontal axis. According to the legend, we plot a distinct line for every choice of $\gamma$. In the right subplot, we show analogously the minimum achieved cost on the vertical axis. Due to the deterministic nature of BeeRS, we run the simulation with every set of distinct $\alpha$ and $\gamma$ exactly once.

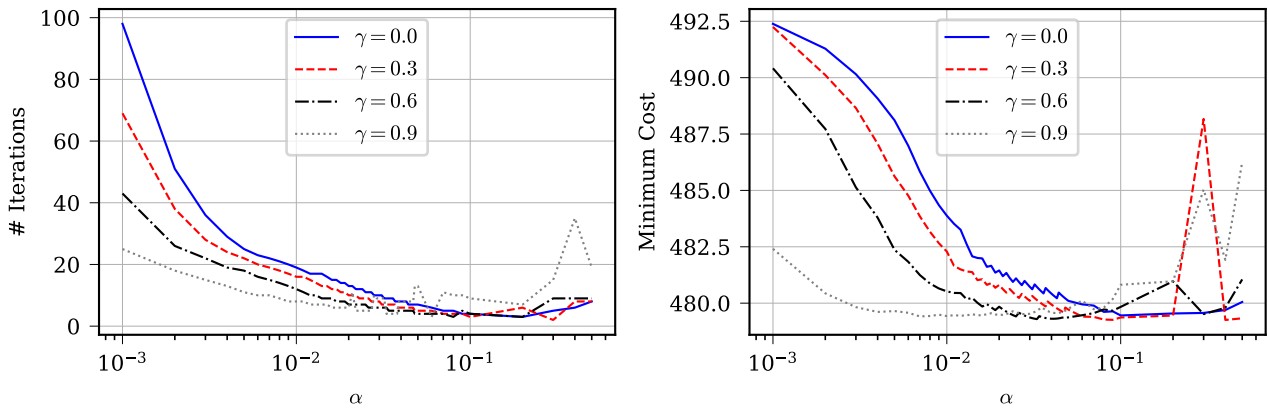

*Figure 6.* We vary the step size $\alpha$ and display the resulting number of iterations (left) and the achieved minimum cost (right) for some momentum parameters $\gamma$.

We observe that, for small values of $\alpha$, we achieve convergence with fewer iterations for larger values of $\gamma$ than for smaller values of $\gamma$. For larger values of $\alpha$, the number of iteration until convergence first gets closer for different values of $\gamma$, and some oscillations begin to appear for $\gamma = 0.9$, and later, the number of iterations until convergence starts to increase again.

Regarding the minimum cost achieved, we observe that for small values of $\alpha$, $\gamma = 0.9$ clearly outperforms all other choices. When $\alpha$ is increased, the performance for other values of $\gamma$ improves quickly, and at a medium value of $\alpha$, $\gamma = 0.6$ achieve a lower cost than $\gamma = 0.9$. For large values of $\alpha$, the minimum achieved cost abruptly rises for some values of $\gamma$, and gradually increases for other values of $\gamma$.

Considering these results, a medium value of $\alpha$, e.g., $\alpha = 0.05$, together with $\gamma = 0.6$ provides a good compromise between the number of iterations until convergence and the achieved minimum cost, and effectively avoids numerical instabilities causing convergence issues for larger values of $\alpha$.

### A.4.1. CHOOSING $\alpha$ AND $\gamma$ FOR DIFFERENT NUMBER OF USERS $n$

While the ablation study in Section A.4 was only carried out for the first $n = 1000$ users of the DBLP dataset, we observed that the resulting hyperparameters $\alpha = 0.05$ and $\gamma = 0.6$ work well for both other problem dimensions $n$ and for the LiveJournal dataset, as long as we consider problem (10). Differently, due to the change of objective in Section 4.3 and the smaller number of variables $m$, the previously used step size yielded insufficient progress. We performed a grid search over the step size and selected the best value, $\alpha = 1000$, while keeping $\gamma = 0.6$ fixed, as determined in the aforementioned ablation study.

## A.5. Further Results of Section 4.2

In addition to the results provided in Section 4.2, we provide in Table 2 the relative suboptimality for the compared approaches. We define the relative suboptimality as

$$\frac{|\varphi - \varphi^\star|}{|\varphi^\star|},$$

where $\varphi^\star$ is the best solution achieved by any of the two approaches.

*Table 2.* Comparison between BeeRS and IPOPT in terms of relative suboptimality.

| NUMBER OF USERS $n$ | 100 | 200 | 300 | 400 | 500 | 1000 | 2000 |
|---|---|---|---|---|---|---|---|
| DBLP | | | | | | | |
| REL. SUBOPT. OF IPOPT | 3.04e-2 | 1.53e-2 | 1.35e-2 | 1.02e-2 | 1.06e-2 | 2.30e-2 | 5.59e-3 |
| REL. SUBOPT. OF BEERS | 0 | 0 | 0 | 0 | 0 | 0 | 0 |
| LIVEJOURNAL | | | | | | | |
| REL. SUBOPT. OF IPOPT | 6.96e-2 | 7.89e-2 | 8.31e-2 | 9.09e-2 | 9.31e-2 | 8.74e-2 | 1.15e-1 |
| REL. SUBOPT. OF BEERS | 0 | 0 | 0 | 0 | 0 | 0 | 0 |

## A.6. Network Administrator Dynamics - Alg. 2

In the following, we repeat the algorithm of (Chitra & Musco, 2020).

---
**Algorithm 2** NAD (Network Administrator Dynamics)

---
1: **while** not converged **do**
2:    $y_k \leftarrow$ For given network weights $w$, compute the equilibrium opinions with (2)
3:    $w_k \leftarrow$ For fixed opinions $y$, solve (11)
4:    $k \leftarrow k + 1$
5: **end while**

---

