# OpenReview forum: "Optimizing Social Network Interventions via Hypergradient-Based Recommender System Design"
_ICML.cc/2025/Conference — ICML 2025 poster_

### Official Review · Reviewer_5iEL · 2025-03-05

**Overall Recommendation:** 3

**Summary:**

The paper studies interventions in the Friedkin-Johnsen opinion formation model. This model is defined by a graph in which each node has a fixed innate opinion and a time-dependent expressed opinion; the model is often used to study polarization and disagreement in networks from a mathematical angle. Many recent works studied interventions in this opinion formation model by considering optimization problems which *either* change the graph topology *or* the innate opinions of the nodes.

The submission's main contribution is to show that one can *simultaneously* optimize for the network topology *and* the opinions. This is done via gradient descent by considering the hypergradient; the main technical advantage here is that this allows to deal with the expressed opinions $y$ which are given by an equation system $Ay=s$ where $s$ is part of the input and $A$ depends on the graph topology.

**Claims And Evidence:**

I think it is mostly fine, but the comparison with the work of Chitra and Musco is inappropriate. See my more detailed comments below.

**Essential References Not Discussed:**

Some more papers could be cited, see my comments below.

**Experimental Designs Or Analyses:**

See below.

**Methods And Evaluation Criteria:**

The definition that was used for polarization in the experiments is non-standard by now. See my comments below.

**Other Comments Or Suggestions:**

* Recent papers in this line of work define the polarization as the variance of the opinions, i.e., $\sum_u (y_u - \bar{y})^2$ where $\bar{y}$ is the average opinion over all nodes. This is not the same as defined in the paper. Sometimes people do consider polarization as $\sum_u y_u^2$ but only if the average opinion is $0$, in which case both notions coincide; here, the assumption of mean-centered opinions is crucial but it is not made in the paper. I would find it more interesting if the paper could show results for this notion of polarization.
* The notation used in the paper is quite non-standard. Typically, the expressed opinions are referred to as $z$ and not as $y$. Also, what is $A$ in the paper is typically referred to as $I+L$ (identity matrix + Laplacian).
* There are quite a few references that could be added. For instance, recent works from Aris Gionis and Sijing Tu gave convergence analysis for gradient descent-based methods in the FJ model and also studied a similar setting with news agencies.

**Other Strengths And Weaknesses:**

Strenghts:
* The theoretical results are interesting and a good contribution to the literature.
* The paper is easy to follow.

Weaknesses:
* The comparison with the Chitra and Musco paper in the experiments is somewhat misleading. The point of the Chitra and Musco paper was not to minimize the overall disagreement; instead, they proposed a model which proceeds in rounds, where the network administrator minimizes the disagreement and then subsequently the opinions converge. Their goal was to study how this particular process might lead to increased polarization. Thus, stating that the algorithm from the paper achieves better objective function values because it can simultaneously minimize over the graph topology and the opinions simultaneously is just not a meaningful comparison.
* Typically, papers in this domain give a model of interventions and then provide optimization algorithms. While the paper does give a quite general optimization approach, it would be better if there were concrete examples in which settings one would expect to optimize over the graph topology and the innate opinions simultaneously. I am also not convinced by the example in Section 4.1.1 because it uses a polarization-metric that is non-standard in the meantime.
* The paper is only evaluated on two datasets, only one of which actually corresponds to a social network. I was surprised that the DBLP dataset (which is a citation network) was used, given that platforms like SNAP make a lot of social network datasets available.
* There is no analysis on how quickly the gradient descent algorithm converges to an optimal solution.

**Update after rebuttal:** The authors have addressed several of my concerns in their rebuttal and thus I have increased my score. I still think that the optimization problems they study could be better motivated, though.

**Questions For Authors:**

* Does your approach also work for the more common notion of polarization that I mentioned above?
* In what kind of model would it make sense that one can simultaneously impact the graph topology and the innate opinions?

**Relation To Broader Scientific Literature:**

I think the main contribution of optimizing opinions and graph topology simultaneously is definitely interesting and something I have not seen before. I appreciate these results. However, in terms of motivation, I think it could be improved (see below).

**Theoretical Claims:**

The theoretical derivations appear to be correct, but I did not check in detail.

---

> ### Author Rebuttal · Authors · 2025-03-31
>
> We thank the reviewer for their feedback and constructive comments.
> First, we would like to clarify one important point.
> Our algorithm optimizes only over the network topology, i.e., the network weights $w$, but not the _innate_ opinions $s$.
> The main optimization problem that we aim to solve, equation (3), has as decision variables
> the weights $w$ and equilibrium _expressed_ opinions $y$.
> Since $y$ is uniquely determined by the constraint (3b), we optimize over $w$ solely.
> We could readily extend our framework to optimize for both $w$ and $s$ simultaneously, as suggested by the reviewer, but
> this is not something that we investigate in this paper.
> We will make sure to clearly elaborate on this point in our revised manuscript.
>
> **Weakness 1** We thank the reviewer for their insightful comment.
> Indeed, the main point of Network Administrator Dynamics (NAD) in Chitra \& Musco, 2020 is to show how minimizing disagreement can have adverse effects on polarization of opinions.  They do have credits for raising this important and critical aspect of this optimization problem, which we will be more careful to acknowledge.
> In this vein, our goal is to  understand the mechanism by which NAD operates and, in turn,
> demonstrate how our algorithm succeeds by using a different mechanism in the same problem setting.
> We believe our comparison to be a meaningful one for two reasons:
>
> 1. The two algorithms operate in the same fashion: they both propose a model which proceeds in rounds, where the network administrator/leader minimizes the disagreement and then subsequently the opinions converge.
>
> 2. our algorithm manages to fix (some of)
> the issues that Chitra \& Musco highlighted with NAD.
> In fact, in their paper they also propose fixing NAD with NAD*, which we also include in our comparison results.
> In essence, we provide a potential solution to one of the main issues raised by Chitra \& Musco.
> We will try to clarify these distinctions and elaborate on these points on the revision of our paper by making sure we give Chitra \& Musco credits for raising a very important and critical aspect of this optimization problem.
>
> **Weakness 2** Regarding the optimization of graph topology and innate opinions, please see our discussion above.
> Regarding the polarization metric, we have rerun our simulations using the metric proposed by the reviewer
> and will include them in the corresponding section.
> We quickly summarize some of them: The reruns of Sections 4.1.2, 4.1.3, and 4.2 with the suggested polarization metric are close to our original results, as can be seen in Figure 1 and 2, and Table 1 on https://imgur.com/a/8GbzUNw.
> Additionally, BeeRS decreases the polarization in Section 4.1.3 even further with the new polarization metric (-44\% compared to -39\%).
>
> **Weakness 3** We thank the reviewer for their useful suggestion.
> We have deployed our algorithm on the soc-LiveJournal1 dataset, which is the largest among the ones available in SNAP, and will include the results in the corresponding section. A preview of the results is available on https://imgur.com/a/8GbzUNw.
> The collected results further confirm the effectiveness of our algorithm.
>
> **Weakness 4** We acknowledge the reviewer's suggestion to analyze the rates of convergence of gradient descent.
> We have performed an empirical analysis of the learning curves of our algorithm, namely objective vs. number of iterations,
> as also suggested by reviewer pBeX; please refer to that response for more information.
> Performing a theoretical analysis can be challenging due to the non-convexity of the problem and could potentially
> require a paper on its own.
> One such analysis, for example, is the classical result in [1, Ch. 1.2] which proves that _vanilla_ gradient descent
> converges to an $\epsilon$-stationary point at a rate of $O(\epsilon^{-2})$, for the case of an unconstrained non-convex smooth optimization
> problem.
> To the best of our knowledge, convergence rates for projected gradient-descent with momentum on a non-convex objective is an open
> problem.
> We consider such theoretical analyses beyond the scope of our paper,
> and reserve the investigation for future work.
>
> **Other Comments or Suggestions 1** Our algorithm can deal with any differentiable objective function, including the one mentioned.
> We collected new results for the proposed metric as mentioned above, with a preview available at https://imgur.com/a/8GbzUNw.
>
> **Other Comments or Suggestions 2** We will highlight this difference in notation.
>
> **Other Comments or Suggestions 3** Thank you for mentioning these works. They are indeed very relevant. We will cite them in the revised manuscript and make an accurate comparison.
>
> **Questions** Please refer to our answers in **Weakness 2** and **Other Comments or Suggestions** for Question 1,
> and to the first paragraph of our response for Question 2.
>
> [1] Nesterov, Yurii. Lectures on convex optimization. Vol. 137. Berlin: Springer, 2018.

---

### Official Review · Reviewer_pBeX · 2025-03-10

**Overall Recommendation:** 3

**Summary:**

This paper investigates opinion polarization in social networks using a large-scale optimization approach to modify network interactions based on the Friedkin-Johnsen model. The authors propose a gradient-based algorithm designed to address this problem in a scalable and computationally efficient manner. Some empirical analysis results are presented to demonstrate the effectiveness.

**Claims And Evidence:**

In general, this paper features a clear motivation of problem definitions and an intuitive proposed method, is generally well written. Assumptions are properly outlined and theoretical derivations are properly provided to motivate the proposed solution. In particular, I like the style that authors offer detailed derivations in Sections 2 and 3, which can offer a clear view on the conditions as well as prerequisites for their contents to be valid.

**Essential References Not Discussed:**

NA

**Experimental Designs Or Analyses:**

Some empirical results are provided to demonstrate the characteristics of the proposed BeeRS method, including running time, GPU-aided computation, and the distribution of edge weights after optimization. However, it seems that the empirical analysis lacks a uniformed metric in terms of the performance, such as the direct illustration of the cost ($\phi$) and the performance tradeoff. In particular, only one real-world dataset DBLP is involved in the experiments, which can be insufficient in terms of validating the empirical effectiveness.

**Methods And Evaluation Criteria:**

The proposed BeeRS algorithm is intuitive and well-motivated, backed by straightforward derivations. On the other hand, as mentioned by the authors in their experiments, "the superior performance of BeeRS comes at the price of an increased runtime", indicating that while the proposed hypergradient method can be time consuming. While this problem can be somewhat alleviated by GPU parallel computing, it could still be a problem given a large number of users or nodes.

**Other Comments Or Suggestions:**

None

**Other Strengths And Weaknesses:**

Please see my comments above in terms of the experiments, in "Experimental Designs Or Analyses". In particular, experiments on additional real-world datasets are encouraged to comprehensively demonstrate the effectiveness of the proposed method.

**Questions For Authors:**

Please refer to my comments in terms of scalability, efficiency, and experiment comprehensiveness. In particular, please see my comments above in terms of the experiments, in "Experimental Designs Or Analyses". In particular, experiments on additional real-world datasets are encouraged to comprehensively demonstrate the effectiveness of the proposed method.

**Relation To Broader Scientific Literature:**

The paper extends existing literature on opinion dynamics and network interventions by formulating social network influence optimization as a scalable hypergradient-based optimization problem under the classical Friedkin-Johnsen model. The proposed hypergradient method can be of independent interest to other audience apart from the Social Sciences community.

**Theoretical Claims:**

The main derivations are in the main body of the paper and I found them to be quite intuitive and clear.

---

> ### Author Rebuttal · Authors · 2025-03-31
>
> We thank the reviewer for the positive comments and their careful reading.
>
> **Methods and Evaluation Criteria** The concern raised on "Methods and Evaluation Criteria" is a valid and important one.
> Indeed, for problems/networks of huge dimensions our algorithm might face computational bottlenecks.
> Nonetheless, our scheme could be adapted and improved in a number of ways to address such practical problems.
> We outline some next:
>
> 1. We could further exploit the capabilities of GPU computation. In Subsection 4.1, we successfully solve a problem
>     with 3 million decision variables on a GPU that currently is not among the state-of-the-art (NVIDIA GeForce RTX 3060 Ti).
>     By using a more performant GPU, or even a cluster of them, we could definitely handle problems of even larger dimensions.
>     In fact, problems with hundreds of millions of decision variables are routinely solved with GPUs in the context of neural network training.
>     This flexibility is a significant advantage of our algorithm, emanating from the fact that it is a simple, first-order method.
>     (Similar to the ones used in neural network training.)
>
> 2. We could reduce the number of weights that we optimize over by employing appropriate heuristics
>     to identify the most important parts of the network,
>     introducing a tradeoff between solution quality and computational requirements.
>     Indeed, our algorithmic approach and our implementation allow the user to specify which
>     weights are immutable and which ones are variable.
>    One meaningful way to choose which weights to optimize (or which areas of the networks to affect) would be to use heuristics to quantify
>   the importance of nodes, and then optimize the weights of edges connected to these nodes.
>   For instance, we could use the degree of a node or, more generally, other centrality measures [1].
> These approaches to choosing which weights to update are especially important since we, typically, do not want
> to modify the weights too much as they affect the users' experience in the network.
>
> 3. We could update only a (randomly chosen) subset of the weights at each iteration, thus giving rise
>     to a stochastic/mini-batch version of our algorithm.
>     This is a GPU-friendly way of addressing problems with huge dimensionality.
>
> [1] Bloch, Francis, Matthew O. Jackson, and Pietro Tebaldi. "Centrality measures in networks." Social Choice and Welfare 61.2 (2023): 413-453.
>
> **Experimental Designs or Analyses/Question**
>
> - We thank the reviewer for their suggestion to include additional datasets in our simulations.
> Please note that in our original manuscript, beside DBPL, we have also tested our algorithm on the real-world Reddit Dataset.
> Further,  we have also deployed our algorithm on the soc-LiveJournal1 dataset,
> and will include the results in our revised manuscript.
> We summarize the results from our additional simulations in Figures 1-3, and Table 1 in https://imgur.com/a/8GbzUNw.
> We highlight that soc-LiveJournal1 is one of the largest directed social networks on the SNAP platform with 4.8 million users and 69 million edges.
>
> - We thank the reviewer for suggesting an empirical analysis with a uniformed metric.
> We include an additional simulation where we study the objective $\varphi$ as a function of the iteratation $k$.
> In particular, we deploy BeeRS on Problem (10) with the polarization metric suggested by Reviewer 5iEL. We use the Reddit, DBLP, and soc-LiveJournal1 datasets, and initialize every simulation i) with initial weights 0, and ii) with 4 additional random initializations. For the total of 5 simulations per dataset we plot the mean cost $\varphi$ with standard deviation as a function of the iteration $k$, as shown in Figure 3 on https://imgur.com/a/8GbzUNw. Please note that there is a large difference in cost between the reddit dataset and the other two datasets, which is explained by the much larger number of users in the latter ones.
>
>     We also rerun the simulations from Sections 4.1.2, 4.1.3, and 4.2 with the polarization metric suggested by Reviewer 5iEL on the additional soc-LiveJournal1 dataset, as can be seen in Figure 1, Table 1, and Figure 2 at https://imgur.com/a/8GbzUNw, respectively.

---

### Official Review · Reviewer_soGD · 2025-03-14

**Overall Recommendation:** 4

**Summary:**

This paper proposes a novel method, Best Intervention for Recommender Systems (BeeRS), which provides a method of using the hypergradient to optimize connection weights in a social network for certain objectives (for example, reducing polarization). The paper models the social network via the Friedkin-Johnson (FJ) model, where each actor has a constant internal opinion and evolving external opinion that is updated and influenced by neighbors in the network. The problem is formulated as bilevel optimization problem, where the lower level models the opinion dynamics among users and the upper level models the recommendation algorithm (termed the “leader”) that adjusts network weights to achieve an objective. Numerical results show an improvement over IPOPT on runtime for the optimization problem. The algorithm is also compared to the Network Administrator Dynamics (NAD) algorithm on real world data from Reddit, and it shows improvement on both the change in polarization and change in disagreement (at the price of an increased runtime).

**Claims And Evidence:**

The claims in the submission are supported by clear evidence. In particular, the experiments are comprehensive and show the improvement of BeeRS over the existing NAD algorithm.

**Essential References Not Discussed:**

I am not aware of any essential references that were not discussed.

**Experimental Designs Or Analyses:**

The experimental designs do seem sound, and I especially appreciate the use of large, real-world datasets for the work.

**Methods And Evaluation Criteria:**

The methods used generally make sense for the problem at hand. I do note some concerns about the practicality of the simulation being conducted (see Other Strengths and Weaknesses below), but overall the methods work well for the problem setup and assumptions.

**Other Comments Or Suggestions:**

N/A

**Other Strengths And Weaknesses:**

The primary strengths of this paper are in the problem formulation and algorithm construction. The paper provides a flexible framework for interventions on social networks. It is also written clearly and has thorough experiments to validate the claims that the authors make regarding the algorithm’s performance.

I think the primary weakness of the paper is a lack of connection back to real-world recommendation systems. While the paper does a good job of using real-world networks from DBLP and Reddit, it is difficult to see how the idealized “leader” in this setup translates to a practical multi-level recommender system that exists out in the wild on a social network. For example, in this work, the leader accomplishes goals by adjusting network weights between users, thereby changing how users influence one another. The paper states that such weights could represent the volume of content from one user that another sees or similar metrics. In practice, social media recommender systems work by ranking content to be seen by a user. How does a ranker system like that “adjust” the weights between users? Does it discount or boost the ranks of certain posts? Does it have to maintain an additional influence model that feeds into the ranking algorithm? This gap is not fatal to the paper, but to me it is a missing piece that makes the difference between a good paper and a great one.

**Questions For Authors:**

1. How would weight adjustments work in a practical real world system? Consider for example a timeline ranking system on a network like Facebook or X.

2. Can the algorithm be further optimized for performance by considering updates in portions of the graph rather than updating the weights of the entire network? Perhaps there is some way to tell which areas of the network will be most influential for the objective overall (e.g. particularly influential users).

**Relation To Broader Scientific Literature:**

This work is a useful extension of existing work on social network interventions, specifically motivated by polarization and disagreement. The advantage of this work comes from the flexibility of the method, given that any differentiable objective can be plugged into the BeeRS algorithm.

**Theoretical Claims:**

I did not verify the proofs of any theoretical claims.

---

> ### Author Rebuttal · Authors · 2025-03-31
>
> We thank the reviewer for their positive evaluation and their constructive comments.
>
> Concerning the weaknesses and questions raised by the reviewer:
>
> **Question/Weakness 1** Thanks for pointing out this aspect. We foresee using the weights as a penalty (/boosting)
> factor in a Weighted PageRank (WPR) [1] or EdgeRank [2] algorithm fashion.
> These algorithms take into account the importance of both the inlinks and the outlinks of the pages and distribute rank scores
> based on the popularity of the pages:
> If the network leader (upper level) decreases the weight of the link from user A to user B then this translates into a penalty
> for all the content shared from A which is a candidate to be visible to B.
> The details of how this would be algorithmically handled, however, are out of the scope of this work and an interesting direction for future research.
>
> **Question 2** We thank the reviewer for their suggestion. Indeed, our algorithmic approach and our implementation allow the user to specify which
> weights are immutable and which ones are variable.
> One meaningful way to choose which weights to optimize (or which areas of the networks to affect) would be to use heuristics to quantify
> the importance of nodes, and then optimize the weights of edges connected to these nodes.
> For instance, we could use the degree of a node or, more generally, other centrality measures [3].
> These approaches to choosing which weights to update are especially important since we, typically, do not want
> to modify the weights too much as they affect the users' experience in the network.
>
> Another interesting way to optimize for performance is to randomly choose which weights to update at each iteration, giving rise
> to a stochastic/mini-batch version of our algorithm.
> This would be particularly helpful when dealing with networks of huge size.
>
> [1] Xing, Wenpu, and Ali Ghorbani. "Weighted pagerank algorithm." Proceedings. Second Annual Conference on Communication Networks and Services Research, IEEE, 2004.
>
> [2] EdgeRank: The Secret Sauce That Makes Facebook's News Feed Tick. techcrunch.com. 2010-04-22. Retrieved 2012-12-08.
>
> [3] Bloch, Francis, Matthew O. Jackson, and Pietro Tebaldi. "Centrality measures in networks." Social Choice and Welfare 61.2 (2023): 413-453.

---

### Official Review · Reviewer_nKxb · 2025-03-25

**Overall Recommendation:** 4

**Summary:**

This paper proposes a gradient descent based method for modifying network weights towards an optimal (general) downstream performance metric, under the framework of Friedkin-Johnsons opinion dynamics. Experiments show significant improvement on computation time on large-scale real-world dataset.

## update after rebuttal
I have read all of the rebuttals, and would like to keep my current rating.

**Claims And Evidence:**

Yes.

**Essential References Not Discussed:**

I feel that the references being cited in the paper are relatively old. There are a few more recent papers working on similar topics  - though none of them are as generalized as this work, but it would be nice to mention them since  it would help further show the great contribution in this paper.

**Experimental Designs Or Analyses:**

Yes.

**Methods And Evaluation Criteria:**

Yes.

**Other Comments Or Suggestions:**

n/a

**Other Strengths And Weaknesses:**

**Strengths**

This is a very nice paper on Friedkin-Johnson opinion dynamics, definitely among the best I have seen in a while. I especially like two things in this paper, the first one not even highlighted by authors themselves:

- It gets rid of the the symmetry assumption made the vast majority of papers working on FJ model. Previous works did so mainly because they needed the symmetry condition to derive the closed-form equilibrium state (i.e. (I+L)^-1 s), which is the basis for many further derivations. This work, however, starts by assuming the network to be directed, and cleverly avoids having to explicitly work with the closed-form solution by denoting it as y*(w). Prop. 2.1 is also good observation.

- It observes and successfully addresses a very important problem, which is that existing studies based on closed-form sensitivity analysis are very computationally expensive, since they usually involves inverting a gigantic matrix. This works elegantly solves this problem with gradient descent. It is also nice to see Assumption 2.2 which generalizes some of the specific performance metrics that people have been studying for years into a more general form.


**Weakness**
- The references being mentioned in "literature review" are mostly before 2020. This might give the readers an impression that the study of FJ model is less active in most recent years, which is not true. For example, [1, 2] involve a similar (albeit narrower) topic of sensitivity analysis on network weights. I encourage the authors to also include these more recent papers, which should further help highlight the broader contribution of this paper.

[1] On the Relationship Between Relevance and Conflict in Online Social Link Recommendations, Wang et al., NeurIPS 2023.

[2] Minimizing Polarization and Disagreement in the Friedkin–Johnsen Model with Unknown Innate Opinions, ArXiv, 2025.

**Questions For Authors:**

n/a

**Relation To Broader Scientific Literature:**

This work has substantial contribution on top of the existing literature.

**Theoretical Claims:**

Not all, but those in Sections 3 and 4.

---

> ### Author Rebuttal · Authors · 2025-03-31
>
> We thank the reviewer for their very positive evaluation and their kind suggestions.
> We will make sure to stress the advantage of considering directed networks in the paper and include the references suggested by the reviewer.

---

### Decision · Program_Chairs · 2025-05-01

**Decision:**

Accept (poster)

**Comment:**

The paper proposes a general and flexible optimization framework in the context of Friedkin-Johnsen opinion dynamics model. The framework allows the use of gradient-based algorithm to solve the optimization tasks, with applications in polarization and other network related tasks. After the rebuttal period, all reviewers are convinced on the contribution of the paper and support the acceptance of the paper. Therefore I recommend the paper be accepted at ICML'2025.

The authors need to seriously address all reviewer comments in their camera-ready version. In particular, I share the concern of one reviewer on the feasibility of intervention in the model: the intervention is by changing the edge weights, but how could it be realized in an actual social platform or recommendation system? More discussions and perhaps examples are needed to support this intervention model.